# Non-uniform Timestep Sampling: Towards Faster Diffusion Model Training

**Tianyi Zheng**[*]
Shanghai Jiao Tong University
Shanghai, China
tyzheng@sjtu.edu.cn

**Cong Geng**
vivo Mobile Communication Co., Ltd
Shanghai, China
gengcong@vivo.com

**Peng-Tao Jiang**
vivo Mobile Communication Co., Ltd
Shanghai, China
pt.jiang@vivo.com

**Ben Wan**
Shanghai Jiao Tong University
Shanghai, China
burn-w@sjtu.edu.cn

**Hao Zhang**
vivo Mobile Communication Co., Ltd
Shanghai, China
haozhang@vivo.com

**Jinwei Chen**
vivo Mobile Communication Co., Ltd
Shanghai, China
jinwei.chen@vivo.com

**Jia Wang**[†]
Shanghai Jiao Tong University
Shanghai, China
jiawang@sjtu.edu.cn

**Bo Li**[†]
vivo Mobile Communication Co., Ltd
Shanghai, China
libra@vivo.com

## Abstract

Diffusion models have garnered significant success in generative tasks, emerging as the predominant model in this domain. Despite their success, the substantial computational resources required for training diffusion models restrict their practical applications. In this paper, we resort to the optimal transport theory to accelerate the training of diffusion models, providing an in-depth analysis of the forward diffusion process. It shows that the upper bound on the Wasserstein distance of the distribution between any two timesteps in the diffusion process is an exponential decrease of the initial distance by a factor of times. This finding suggests that the state distribution of the diffusion model has a non-uniform rate of change at different points in time, thus highlighting the different importance of the diffusion timestep. To this end, we propose a novel non-uniform timestep sampling method based on the Bernoulli distribution, which favors more frequent sampling in significant timestep intervals. The key idea is to make the model focus on timesteps with larger differences, thus accelerating the training of the diffusion model. Experiments on benchmark datasets reveal that the proposed method significantly reduces the computational overhead while improving the quality of the generated images.

## CCS Concepts

• **Computing methodologies → Maximum likelihood modeling**; **Reconstruction**.

---

[*]This work was done during Tianyi Zheng's internship at vivo.
[†]Corresponding Authors.

---

## Keywords

Diffusion Model, Optimal Transport, Wasserstein distance.

**ACM Reference Format:**
Tianyi Zheng, Cong Geng, Peng-Tao Jiang, Ben Wan, Hao Zhang, Jinwei Chen, Jia Wang, and Bo Li. 2024. Non-uniform Timestep Sampling: Towards Faster Diffusion Model Training. In *Proceedings of the 32nd ACM International Conference on Multimedia (MM '24), October 28-November 1, 2024, Melbourne, VIC, Australia.* ACM, New York, NY, USA, 10 pages. https://doi.org/10.1145/3664647.3680912

## 1 Introduction

In recent times, generative models, particularly diffusion-based generative models [6, 25, 46, 57], have garnered significant attention for their notable achievements in computer vision [4, 35, 37, 51], natural language processing [3, 47, 54], temporal data modeling [1, 5, 12, 44] and AI Security [13, 14, 36]. The diffusion model comprises two primary stages: forward diffusion and reverse diffusion. In the forward diffusion stage, the objective is to facilitate the transformation of the original distribution into a Standard Gaussian distribution. On the other hand, the reverse diffusion stage aims to transition from a Standard Gaussian distribution back to the data distribution, thereby accomplishing the generation of new samples.

Despite the significant achievements of diffusion models as evidenced by several studies [25, 41, 48], the considerable computational demands required for their training restrict their utility in a broad spectrum of applications. ADM [10] presents an enhanced adaptive group normalization framework that more effectively integrates temporal data within residual blocks, thereby accelerating the training process of diffusion models. EDM [28] designs advanced noise schedules to improve training efficiency. Meanwhile, many studies propose the various re-weighted loss functions, effectively enhancing the training efficiency of diffusion models. For instance, the P2-weight method [7] focuses on images with medium signal-to-noise (SNR) ratios, believed to contain rich semantic information. To leverage this, P2-weight introduces a re-weighted loss function specifically targeting these medium SNR ratios. This approach aims

to enhance the model's ability to learn visual concepts, thereby boosting training efficiency. Moreover, Min-SNR [23] and ANT [20] discover that the optimal weight gradients for different noise distributions conflict, leading to a notable slowdown in the training of diffusion models. Therefore, Min-SNR proposes a clamped SNR-weighted loss function to reduce this conflict and accelerate the training process. ANT [20] proposes an Uncertainty Weighting strategy [30] to faster training the diffusion model. Similarly, the Debias [56] methods also design a re-weighted loss function based on SNR to reduce the bias of different noise distributions and speed up the training. While these methods somewhat expedite the training process of the diffusion model, they all employ uniform timestep sampling methods in the training stage. *Our subsequent theoretical analysis suggests that uniform timestep sampling methods may be sub-optimal for the training of diffusion models, primarily because the variability across different noise distributions in the forward diffusion is not uniform.*

In this paper, we introduce an approach that leverages optimal transport (OT) theory to examine the forward diffusion process of the diffusion model. OT theory provides enlightenment to the changes in distance between different distributions in the Wasserstein space. More importantly, the forward process of the diffusion model can be modeled as a gradient flow in the Wasserstein space, which provides a good understanding of the change in distribution of the diffusion process. As depicted in Figure 1a, there is a notable exponential diminution in the Wasserstein distances concomitant with the advancement of the forward diffusion process. This analysis confirms that the upper bound of the Wasserstein distance between any two specified moments exhibits an exponential reduction relative to their initial disparity. Figure 1b presents empirical validation of this conclusion. Based on this foundation, we propose a novel non-uniform timestep sampling method named Bernoulli Distribution-Based Sampling for training diffusion models. We call our method Denoising Diffusion Probabilistic Models with Bernoulli Sampling (DDPM-BS). This design concentrates on diffusion stages with distinct distribution differences, enhancing training efficiency and improving the generative performance of diffusion models. Furthermore, our approach is compatible with existing enhancement methods (e.g., P2-weight, EDM, etc.), allowing for seamless integration that can leverage the advantages of various designs to further elevate training efficiency and generative quality. Our contributions can be summarized as follows:

- We employ optimal transport theory to analyze the distributions at various timesteps of the forward diffusion process from different initial distributions. This analysis concludes that the upper bound of the Wasserstein distance between different distributions decreases exponentially with time, guiding improvements in diffusion model training.
- We design a novel non-uniform training timestep sampling method based on Bernoulli distribution, significantly speeding up diffusion model training and enhancing generation quality. Furthermore, our design is also beneficial to other improved designs of the diffusion model.
- Extensive experiments across various benchmark datasets demonstrate that our DDPM-BS method can significantly

expedite the diffusion model training and improve the generation quality.

## 2 Related Work

### 2.1 Diffusion-Based Generative Models.

Diffusion models are proposed by [48] and improved by [17, 25, 41]. Recently, the ADM [10] can generate higher-quality images than Generative Adversarial Networks (GANs) [21, 22, 52]. EDM [28] enhances training efficiency and sample quality through advanced noise schedules and network architectures. However, training a diffusion model like ADM and EDM needs substantial computational cost, thereby constraining the application of these models. To reduce the computational cost, the P2-Weight [7] method designs the re-weighted loss functions to speed up the training. Meanwhile, Min-SNR [23] and ANT [20] analyze the training process of diffusion models through a multi-task learning perspective. Therefore, they propose different re-weighted loss functions to speed up the training. Furthermore, E-TSDM [55] find that Lipschitz singularities pose a threat to the stability of the training. Therefore, E-TSDM shares the timestep with large Lipschitz constants to reduce the instabilities in the training. However, the uniform timestep sampling approach adopted by these methods during the training stage may not be optimal, potentially impacting training efficiency and the quality of the generated samples. Furthermore, these enhanced designs are orthogonal to our timestep sampling method, indicating that they can be integrated with our approach without conflict.

### 2.2 Optimal Transport.

The Wasserstein distance in optimal transport theory is a distance function defined between probability distributions on a given metric space. The 2-Wasserstein distance between two probability measures $\mu, \nu$ is:

$$W_2(\mu, \nu) := \inf \left\{ \int_{\mathbb{R}^d \times \mathbb{R}^d} \|x - y\|^2 d\gamma : \gamma \in \Pi(\mu, \nu) \right\}^{\frac{1}{2}}.$$

Here, $\Pi(\mu, \nu)$ represents the set of all joint distributions (couplings) on $\mathbb{R}^d \times \mathbb{R}^d$ that have $\mu$ and $\nu$ as their respective marginals.

The optimal transport theory and stochastic differential equations (SDE) are closely related [15]. Therefore, the Wasserstein distance has been utilized to explain the diffusion-based generative model [18, 19, 31, 33] in several works. Moreover, DPM-OT [34] accelerates inference speed during generation by solving a semidiscrete OT problem. Even though these studies provide valuable theoretical insights, they have yet to effectively translate these findings into tangible improvements in the training of diffusion models. In this paper, we not only provide an in-depth theoretical analysis of the diffusion model, but also propose an improved strategy for training, which greatly improves the training speed and generation quality of the diffusion model.

## 3 Method

In this section, we provide an introduction to the background of the diffusion model and the optimal transport theory in Section 3.1. Subsequently, in Section 3.2, we explore the analysis of the diffusion process using the Wasserstein gradient flow and experimental

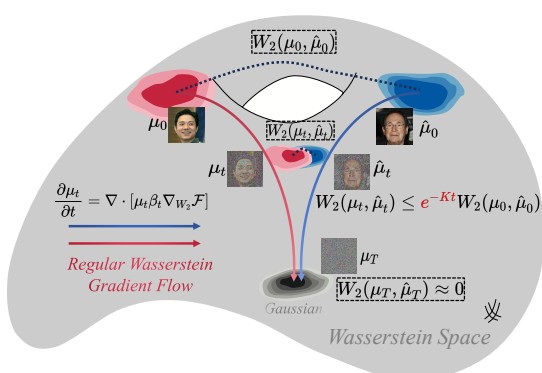

**(a) The forward diffusion process in the Wasserstein space.**

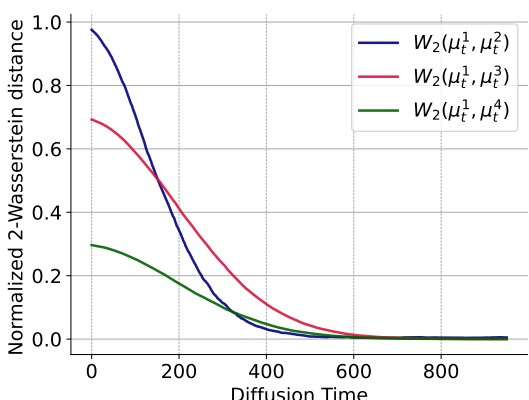

**(b) The Wasserstein distance varies with the diffusion process.**

**Figure 1: Illustration of a Wasserstein gradient flow and experimental results.**

verification of the conclusion. Finally, in Section 3.3, we present our timestep sampling method based on theoretical insights.

## 3.1 Background

The aim of the forward diffusion process is to transform the complex image distribution $q(x_0)$ into simple normal Gaussian distribution $\mathcal{N}(0, I)$. In the forward diffusion process, we define the noise schedule $\beta_t$, Meanwhile, with $\alpha_t := 1 - \beta_t$ and $\bar{\alpha}_t := \prod_{s=0}^{t} \alpha_s$ we assume in each time $t$, the probability density function of $q(x_t|x_0)$ is

$$q\left(x_t \mid x_0\right) = \mathcal{N}\left(x_t; \sqrt{\bar{\alpha}_t} x_0, \left(1 - \bar{\alpha}_t\right) I\right)$$

$$x_t = \sqrt{\bar{\alpha}_t} x_0 + \sqrt{1 - \bar{\alpha}_t} \epsilon. \quad (1)$$

In the training stage, we train a model $\epsilon_\theta(x_t, t)$ to predict the added noise $\epsilon$ in each timestep.

In the inference stage, we start from a known distribution, such as the standard Gaussian distribution, and reverse each step of the noise-corrupted latent $x_{t-1}$ from $x_t$. However, the distribution $q(x_{t-1}|x_t)$ is based on the entire data distribution. Therefore, in DDPM, a neural network parameterized by $\theta$ is employed to estimate $q(x_{t-1}|x_t)$ using the following equation:

$$p_\theta\left(\boldsymbol{x}_{t-1} \mid \boldsymbol{x}_t\right) := \mathcal{N}\left(\boldsymbol{x}_{t-1}; \mu_\theta\left(\boldsymbol{x}_t, t\right), \Sigma_\theta\left(\boldsymbol{x}_t, t\right)\right).$$

Leveraging Bayes' rule, we can use the trained diffusion model to predict the previous denoising distribution step by step, employing the following equation:

$$x_{t-1} = \frac{1}{\sqrt{1 - \beta_t}} \left(x_t - \frac{\beta_t}{\sqrt{1 - \alpha_t}} \epsilon_\theta\left(x_t, t\right)\right) + \sigma_t z.$$

When we make time tend to be continuous [50], the generalized forward diffusion process can be expressed as stochastic differential equations (SDEs) form:

$$dx = f(x, t)dt + g(t)dw. \quad (2)$$

The SDE 2 in Euclidean space describes the change of probability distribution over time in the forward diffusion process, while each distribution in Euclidean space corresponds to an element in the Wasserstein space. In order to gain a better understanding of the

different distributions in the forward diffusion process, we can shift our attention to the Wasserstein space. Indeed, the Wasserstein gradient flow elucidates the dynamics of probability density as it follows the steepest descent direction of the functional $\mathcal{F}$ (e.g. KL divergence), relative to the Riemannian metric induced by the 2-Wasserstein distance [2, 11]. The Wasserstein gradient can be expressed in the following form [2], where $\mathcal{F}'(\mu)$ represents the variance of $\mathcal{F}$ with respect to the measure $\mu$.

$$\nabla_{W_2} \mathcal{F} = \nabla \mathcal{F}'(\mu).$$

The Wasserstein gradient of the functional $\mathrm{KL}(.\|\pi)$ at $\mu$ is

$$\nabla_{W_2} \mathrm{KL}(.\|\pi) = \nabla \log\left(\frac{\mu}{\pi}\right).$$

Then using the functional $\mathrm{KL}(.\|\pi)$ and the continuity equation, we obtain the equation of the Wasserstein gradient flow as:

$$\frac{\partial \mu_t}{\partial t} = \nabla \cdot \left[\mu_t \nabla \log\left(\frac{\mu_t}{\pi}\right)\right].$$

## 3.2 Theoretical Insight into Forward Diffusion via Wasserstein Gradient Flow

Wasserstein gradient flow is a differential equation of probability measures. When we scale a time-varying $\beta_t$ at time $t$, we can get the regular gradient flow [19]. The regular gradient flow satisfies the following continuity equation:

$$\frac{\partial \mu_t}{\partial t} = \nabla \cdot \left[\mu_t \beta_t \nabla_{W_2} \mathcal{F}\right]. \quad (3)$$

In the regular gradient flow, the velocity vector is defined as $v_t = -\beta_t \nabla_{W_2} \mathcal{F}$.

Then we can prove that the corresponding Fokker-Planck equation [45] of the SDE 2 is equivalent to the regular Wasserstein gradient flow [19] when the functional on the Wasserstein space is defined by $\mathcal{F}(\mu) = \mathrm{KL}(.\|\pi)$. The corresponding Fokker-Planck equation [45] is:

$$\frac{\partial \mu_t}{\partial t} = -\nabla \cdot (\mu_t f) + \frac{1}{2} \nabla \cdot \left(\nabla \left(g_t^2 \mu_t\right)\right). \quad (4)$$

Based on this relationship, we have the following lemma.

**Lemma 1.** Consider a regular Wasserstein gradient flow, as defined in Equation 3, initiating from a data distribution $\mu_0$ and converging to a normal Gaussian distribution $\mu_T = \mathcal{N}(0, I)$. With the selection of $f = \beta_t x$ and $g_t = \sqrt{2\beta_t}$, the family of measures $\{\mu_t\}_{t=0}^T$ derived from the Fokker-Planck equation 4 is equivalent to the family of measures corresponding to this gradient flow.

The detailed proof of Lemma 1 is in the Appendix Section 2.1. $\mu_t$ is the noisy data distribution in the forward diffusion at time $t$. This leads us to the following remark.

**Proposition 1.** When set the $f = \beta_t x$ and $g_t = \sqrt{2\beta_t}$, the SDE 2 can be written as:

$$dx = -\beta_t x dt + \sqrt{2\beta_t} dw. \tag{5}$$

This SDE describes the forward diffusion process of the diffusion model [25, 50].

Proposition 1 implies that the forward diffusion process of diffusion model [25, 50] can be equivalently thought as a regular Wasserstein gradient flow [19]. Meanwhile, at any time $t$, the constant $\beta_t$ corresponding to different starting distributions is the same. Therefore, based on the solid mathematical framework of Wasserstein gradient flow [2, 26, 53], We can analyze the forward diffusion process based on the properties of Wasserstein gradient flow in the Wasserstein space. Therefore, we have the following theorem.

**Theorem 1.** Consider two distinct initial distributions $\mu_0$ and $\hat{\mu}_0$ on the data manifold $M$, which is equipped with a reference measure $v = e^{-V}$ vol, satisfying $Hess_\mu \geq K$. Let $\mu_t$ and $\hat{\mu}_t$ represent the distributions at time $t$ in the forward diffusion process described in Proposition 1, originating from $\mu_0$ and $\hat{\mu}_0$ respectively. For all $t > 0$, the following inequality holds

$$W_2\left(\mu_t, \hat{\mu}_t\right) \leq e^{-Kt} W_2\left(\mu_0, \hat{\mu}_0\right). \tag{6}$$

This theorem can be proved in terms of Riemannian geometric (in pure Otto's formalism), and the detailed proof procedure is in the Appendix Section 2.2. The determination of a precise value for $K$, denoting the lower bound of the curvature on the data manifold $M$ [15], presents a significant challenge. Despite this, Theorem 1 elucidates critical theoretical insights on the forward diffusion mechanism inherent in the diffusion model. During the forward diffusion stage, the upper bound of Wasserstein distances between different initial distributions declines rapidly over time. This means that the different initial distributions become very similar at an exponential rate. Then we have the following proposition.

**Proposition 2.** Based on Theorem 1, we observe that during the forward diffusion process, the upper bound of the 2-Wasserstein distance between different distributions exhibits an exponential decrease over time. For a sufficiently small $\delta$, when $t > \frac{\ln \delta}{|K|}$, the following inequality holds:

$$W_2(\mu_t, \hat{\mu}_t) \leq \delta \cdot W_2(\mu_0, \hat{\mu}_0). \tag{7}$$

This proposition tells us that when diffusion has proceeded to a certain point, the 2-Wasserstein distance between the different distributions is already almost 0. The distributions at this point are

virtually the same. Time intervals before this point can be considered significant intervals in the diffusion stage and this provides a theoretical basis for designing improved training methods for diffusion models.

Then, we use the entropic regularization algorithm [9, 16] to empirically evaluate the Wasserstein distance between different distributions in the forward diffusion process. We select different images in the ImageNet dataset as a distribution. Then we use Equation 8 to empirically evaluate the 2-Wasserstein distance between different distributions by setting $C(x, y) = \|x - y\|^2$. This algorithm can compute the 2-Wasserstein distance between different distributions efficiently.

$$W_2(\mu_t, \hat{\mu}_t) \approx \min_{\pi_1 = \mu_t, \pi_2 = \hat{\mu}_t} \int_{\chi^2} C d\pi + \varepsilon KL(\pi \mid \mu_t \otimes \hat{\mu}_t). \tag{8}$$

As illustrated in Figure 1b, we observe an exponential decrease in the 2-Wasserstein distance among various distributions. Initially distinct due to their unique characteristics, these distributions become more similar as the forward diffusion process progresses. This finding challenges the efficacy of uniform timestep sampling in training diffusion models, suggesting that a more targeted timestep sampling approach increasing sampling density at timesteps with notable distributional shifts could better.

## 3.3 Bernoulli Distribution-Based Time Sampling Method

In this section, we propose the Denoising Diffusion Probabilistic Models with Bernoulli Sampling (DDPM-BS) inspired by our analysis of optimal transport theory. Based on the Theorem 1 and Proposition 2, we can divide the timesteps of the diffusion process into significant timestep intervals. Within the significant timestep intervals, notable disparities are observed among the various distributions. Conversely, in the remaining timestep intervals, these disparities diminish, leading to smaller differences. Therefore, we introduce a non-uniform time sampling method that focuses more on significant distributional time intervals. This approach is designed to ensure the model is effectively trained to manage diffusion processes with notable distributional differences.

Specifically, we use $t_\delta$ to represent the threshold value of significant timestep intervals. Consequently, we partition the set of intervals $t_1, t_2, ..., t_\delta, ..., t_T$ into two subsets: $t_1, ...t_\delta$ and $t_\delta, ..., t_T$. Our strategy samples with probability $p$ in significant intervals and $1 - p$ otherwise. This approach yields the following proposition.

**Proposition 3.** During the training stage, let $X_n$ be a random variable representing a sample at time $t_n$ in the timestep series $\{t_1, t_2, \ldots, t_\delta, \ldots, t_T\}$. Specifically, $Pr[X_n = 1]$ denotes the probability of sampling within the significant intervals at time n. The probability of having exactly $k$ samples in the significant intervals, denoted by $Pr[S_n = k]$, is given by the binomial formula $\binom{n}{k} p^k (1 - p)^{n-k}$, consistent with a Bernoulli distribution.

This proposition elucidates the probability density function (PDF) for sampling within significant intervals. Our non-uniform timestep sampling method, following the Bernoulli distribution, is thus named Bernoulli Distribution-Based Sampling. Moreover, we maintain uniform sampling in each sub-interval, both significant and

non-significant. Therefore, the sampling weight of our method in each timestep is non-uniform and can be written as a Bernoulli distribution:

$$f(t) = \begin{cases} p & \text{for } t \in [0, t_\delta] \\ 1 - p & \text{for } t \in [t_\delta, t_T] \end{cases}.$$

The proposed Bernoulli Distribution-Based Sampling training algorithm for the diffusion model is presented in Algorithm 1. Here, $\mathbb{U}$ denotes a uniform distribution. It is important to note that accurately estimating $t_\delta$ is often challenging. Further discussion about the choice of $t_\delta$ is available in Section 4.4. In all our experiments, we use $t_\delta/T = 0.8$ to classify significant time sub-intervals. Then we present a comparison plot of various sampling methods in Figure 2. It becomes evident that we can regulate the sampling frequency during the training stage within different intervals by adjusting the parameter $p$ for the Bernoulli distribution. Specifically, setting $p > 0.5$ allows us to focus timestep sampling more intensively on the significant time sub-intervals. Therefore, the corresponding training loss of DDPM-BS can be written as

$$\mathcal{L}(\epsilon_\theta) := \mathbb{E}_{t \sim \mathcal{B}_\mathcal{U}(0,T), \mathbf{x}_0 \sim q(\mathbf{x}_0)} \left[ \|\epsilon_\theta (\alpha_t \mathbf{x}_0 + \sigma_t \epsilon, f_\mathbb{T}(t)) - \epsilon\|_2^2 \right] \quad (9)$$

We represent our timestep sampling strategy with $\mathcal{B}_\mathcal{U}(0, T)$, and denote $\epsilon$ as a random variable following the standard Gaussian distribution $\mathcal{N}(0, \mathbf{I})$. Furthermore, DDPM-BS only adjusts the timestep sampling strategy, indicating it is orthogonal to other improvement strategies. Therefore, it can easily be integrated with these strategies for enhanced performance of the diffusion model.

Beyond our Bernoulli distribution-based non-uniform timestep sampling method, we quantitatively evaluate other distribution-based methods in Section 3.4 in the Appendix. Experimental results indicate that similarly focusing on significant time intervals in other non-uniform sampling methods also boosts training speed and enhances generate quality.

---

**Algorithm 1** Bernoulli Distribution-Based Sampling

---

**Require:** Upper bound $\delta$, Bernoulli parameter $p$.
 1: Estimating the significant interval $t_\delta$ using Eq. 8
 2: **repeat**
 3:     $x_0 \sim q(x_0)$
 4:     Generating random numbers $u \sim \mathbb{U}[0, 1]$
 5:     **if** $u < p$ **then**
 6:         $t \sim \mathbb{U}(\{1, \ldots, t_\delta\}), \epsilon \sim \mathcal{N}(0, I)$
 7:     **else**
 8:         $t \sim \mathbb{U}(\{t_\delta, \ldots, t_T\}), \epsilon \sim \mathcal{N}(0, I)$
 9:     **end if**
 10:   compute $x_t$ using Eq. 1
 11:   take a gradient descent step on $\nabla_\theta \|\epsilon - \epsilon_\theta (x_t, t)\|^2$
 12: **until** converged

---

# 4 Experiment

## 4.1 Experiment Setup

**Models and Hyperparameters.** DDPM-BS employs the same model architecture as ADM [10]. Since the U-Net of ADM is more

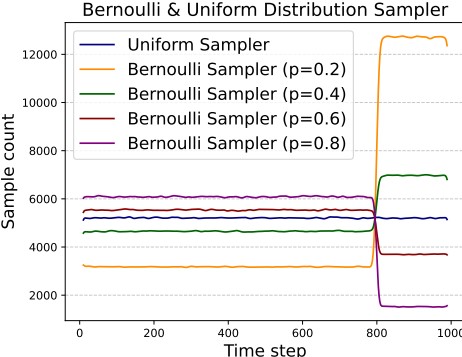

**Figure 2: Comparison of Various Sampling Methods.**

advanced than DDPM. Throughout the training stage, we maintain $T = 1,000$ for all experiments. We conduct comparative experiments using the CIFAR-10 (32×32), ImageNet (32×32), CelebA (64×64) [38], Stanford Cars (128×128) [32] FFHQ (128×128) [29], CelebAHQ (256×256) [27] and AFHQ-D (256×256) [8] datasets for unconditional image generation. More details about our experiment setup are shown in Section 4 in the Appendix.

**Evaluation.** We adopt the widely-used Frechet Inception Distance score (FID) [24] and sFID [40] to evaluate the sample quality. When comparing against previous methods, we follow the previous work [25, 41] and generate $50K$ samples for each trained model on CIFAR-10, ImageNet, CelebA and FFHQ. Meanwhile, we generate $10K$ samples for Stanford Cars, CelebAHQ and AFHQ-D since the size of training data is limited. We utilize the full training set to compute the reference distribution statistics for all datasets.

## 4.2 Quantitative Comparison

**Comparison with DDPMs.** Figures 3 offers a quantitative comparison between ADM-BS and ADM on the different datasets. The results demonstrate that ADM-BS outperforms ADM across all these datasets. Additionally, it is noteworthy that ADM-BS achieves equivalent performance to ADM even before reaching convergence. For example, on CIFAR-10, ADM-BS achieves a 2.22 × faster rate in matching ADM's FID score. Meanwhile, for CelebA and FFHQ datasets, ADM-BS achieves accelerations of 1.47 × and 4.67 × respectively to obtain the same FID as ADM. Meanwhile, for the higher resolution CelebA-HQ and AFHQ-D datasets, ADM-BS reaches a rate that is 3.57 × and 1.90× respectively faster in matching ADM's FID score. Furthermore, our ADM-BS also achieves accelerations of 2.29 × on the large-scale dataset ImageNet.

**Comparison with previous state-of-the-art methods.** We conduct a comparative analysis of diffusion models trained using our method against various previous state-of-the-art models [7, 23, 41, 56] on the CIFAR-10 (32 × 32), FFHQ (128 × 128) and AFHQ-D (256 × 256) datasets, exploring different sampling step settings $T$. For a fair comparison, we employed the same model architecture and focused on the best-performing iterations of the different methods. Moreover, we provide the FID scores concerning the number of training iterations of each method in the Appendix Section 3.1. Additionally, we assess the impact of varying sampling steps, utilizing the

**Table 1: Quantitative comparison. Comparison of FID across various state-of-the-art models on different benchmark datasets.**

| Dataset | Inference Steps $T'$ | iDDPM | ADM | P2-Weight | Min-SNR | Debias | ADM-BS |
|---------|---------------------|-------|-----|-----------|---------|--------|--------|
| CIFAR-10 | 100 | 3.96 | 3.47 | 3.42 | 3.43 | 3.36 | **3.30** |
| | 300 | 3.64 | 2.97 | 3.11 | 3.18 | 3.06 | **2.81** |
| | 1000 | 3.31 | 3.01 | 3.19 | 3.23 | 3.17 | **2.90** |
| FFHQ | 50 | 21.08 | 18.14 | 13.56 | 14.19 | 13.05 | **12.19** |
| | 100 | 16.27 | 14.07 | 11.96 | 12.46 | 11.86 | **10.73** |
| AFHQ-D | 50 | 23.13 | 19.72 | 18.09 | 19.02 | 18.82 | **17.06** |
| | 100 | 20.44 | 17.21 | 16.18 | 17.03 | 15.91 | **14.08** |

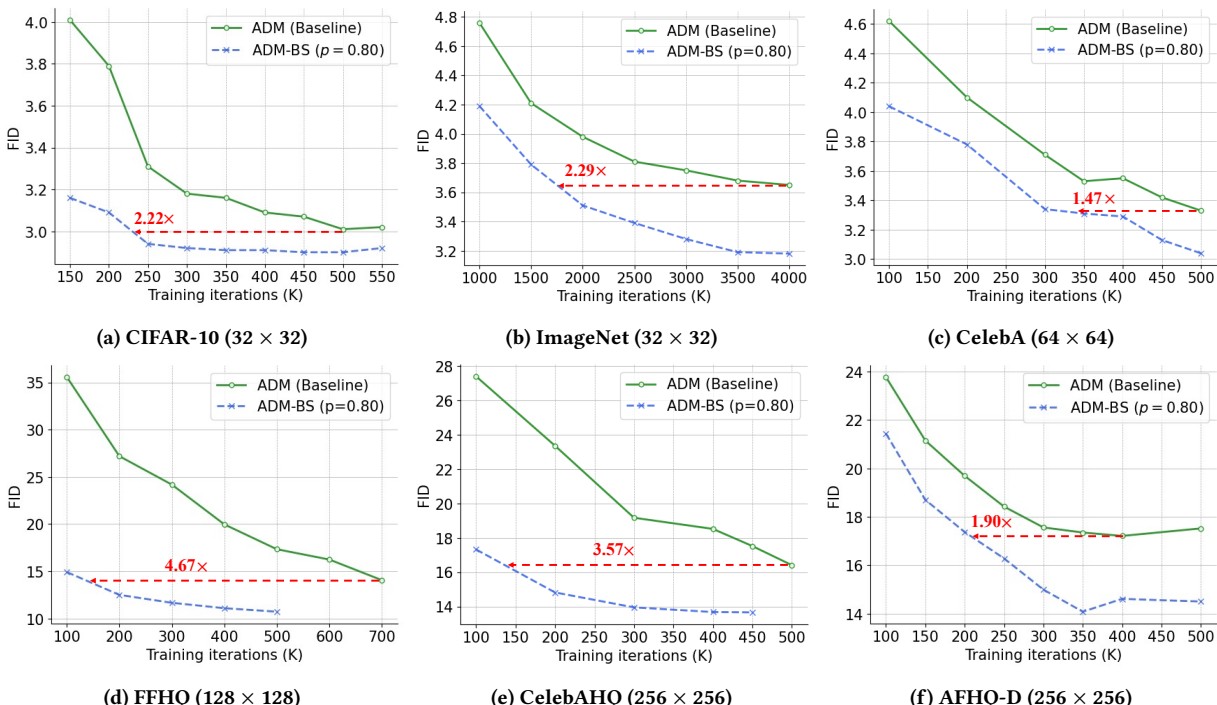

**Figure 3: FID scores concerning the number of training iterations on various datasets.**

respacing technique [10] for step reduction. The results presented in Table 1, indicate that our method consistently outperforms others across multiple datasets and sampling step configurations. Notably, certain weighting strategies (e.g., Min-SNR [23], Debias [56]) do not perform as well as the baseline model under some sampling step scenarios in the low-resolution CIFAR-10 dataset, suggesting a lack of robustness. In contrast, our method demonstrated superior performance under all tested conditions and datasets.

**Integrating with other improvement method.** Our non-uniform timestep sampling strategy is designed to not conflict with other training improvement methods for diffusion models. Because of this compatibility, we attempt to combine our method with previous methods, such as the Min-SNR [23] and P2-Weight [7]. We apply this combination in the training on the AFHQ-D dataset. Table 2

shows significant improvements in FID and sFID metrics. This suggests that combining our method with other training improvement algorithms can further boost model performance, highlighting our method's generalizability. Moreover, our ADM-BS also can be integrated with the Input Perturbation (IP) method [43] to mitigate the bias issue in the diffusion model. For detailed experimental details and results, please refer to 3.3 in the Appendix.

**Fine-Grained Data.** While commonly used datasets such as ImageNet, FFHQ, and AFHQ-D are prevalent in generative modeling studies, they often lack the expressive power required to capture extremely fine-grained differences. These standard datasets primarily serve the purpose of broader category identification, rather than emphasizing fine-grained distinctions within categories. Given that the intricate details hold significance for generation applications,

**Table 2: Experimental results combining our method with other improved methods on AFHQ-D datasets.**

| Method | P2-Weight | + BS | Min-SNR | + BS |
|--------|-----------|------|---------|------|
| FID | 16.18 | **13.34** | 17.03 | **13.89** |
| sFID | 47.88 | **46.81** | 47.94 | **46.79** |

we evaluate the capabilities of ADM-BS in handling fine-grained details on Stanford Cars datasets. The results illustrated in Figure 4 reveal that ADM-BS achieved about 1.51× acceleration in attaining an equivalent FID score compared to the ADM. Furthermore, it is observed that ADM-BS consistently outperforms ADM in terms of generation quality.

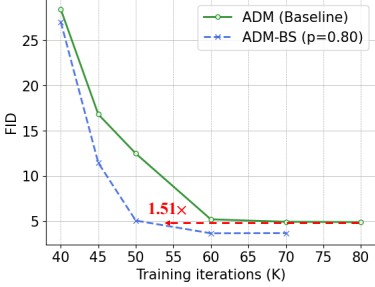

**Figure 4: FID scores concerning the number of training iterations on Fine-Grained Stanford Cars datasets.**

## 4.3 Fast Sampling

The extensive inference process of diffusion models restricts their practical applications, prompting the development of various accelerated inference algorithms, such as DDIM [49], Epsilon Scaling [42], and DPM-Solver [39]. These methods have significantly enhanced the inference efficiency of diffusion models. Consequently, it is essential to evaluate the compatibility and performance of our model in conjunction with these accelerated approaches. As depicted in Table 3, ADM-BS surpasses the baseline in terms of performance under rapid inference conditions. Additionally, our approach demonstrates minimal variation across different sampling steps, indicating enhanced stability and robustness of ADM-BS.

**Table 3: Comparison between ADM and ADM-BS in different fast samplers on AFHQ-D datasets. We use the FID-10K as the evaluation metric which is the same as the previous experiments. NFE means the number of function evaluations.**

| Fast Samplers | DDIM | | Epsilon Scaling | | DPM-Solver | |
|---------------|------|------|-----------------|------|------------|------|
| NFE | 25 | 50 | 25 | 50 | 20 | 50 |
| ADM | 23.63 | 18.50 | 24.22 | 19.52 | 18.49 | 17.49 |
| ADM-BS | 20.35 | 16.59 | 20.29 | 15.35 | 16.48 | 16.17 |

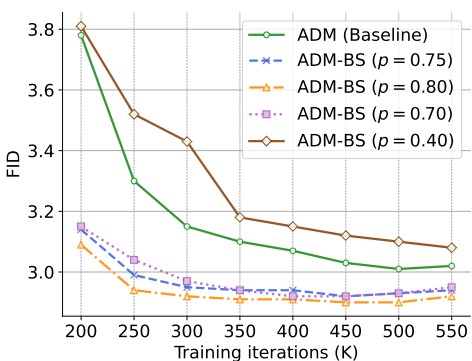

**Figure 5: FID scores with respect to the Bernoulli distribution parameter $p$ on CIFAR-10 dataset.**

## 4.4 Ablation Study

**Bernoulli distribution parameter $p$.** As the parameter $p$ governs the probability of sampling at significant intervals, it directly influences the sampling frequency at these intervals during training. Therefore, we conduct ablation experiments to investigate its impact. The results are shown in Figure 5. We observe that as the parameter $p$ increases, the model converges at an accelerated rate. When $p$ reaches 0.8, the model achieves an FID score of less than 3 within 250K training iterations, outperforming the final converged results of the ADM method. This phenomenon is attributed to ADM-BS, which places greater emphasis on intervals characterized by significant differences. This effectively expedites convergence and enables the model to pay more attention to distribution disparities, resulting in improved generative quality. Additionally, we note that setting $p$ below 0.5 leads to slower convergence compared to the ADM method. By the time the number of training iterations reaches 500K, the ADM has essentially converged, whereas ADM-BS has not reached full convergence at that training iteration. For more discussion about $p$, please refer to 3.2 in the Appendix.

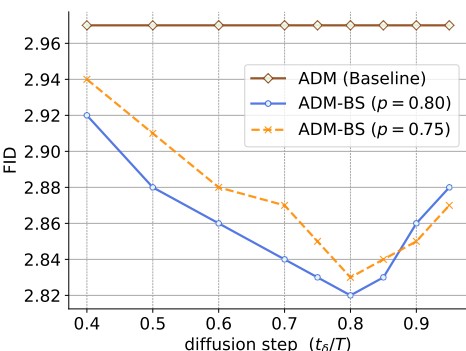

**Figure 6: FID scores with respect to different choice of $t_\delta/T$.**

**The upper bound $\delta$.** The parameter $\delta$ plays a pivotal role in defining the range of significant intervals within a distribution. This determination is based on the premise that when the Wasserstein distance between two distributions is less than $\delta$ times their initial

values, the distributions are considered markedly similar. However, the computation of the Wasserstein distance is very difficult and complex, although we empirically validate the decreasing trend of the Wasserstein distance in Figure. 1b, there is a bias in this estimation and only serves primarily as a motivational figure. Intuitively, we think $\delta$ should be a relatively small value because this implies a more similar distribution. In our experiments, we substitute $\delta$ with various $t_\delta/T$ ratios and evaluate the model's performance across these different ratios, as illustrated in Figure 6. We observe that optimal performance consistently occurs at a $t_\delta/T$ ratio of approximately 0.8 for various values of $p$. Based on this finding, we select $t_\delta/T = 0.8$ as the significant interval threshold for all subsequent experiments. Notably, the model's performance with any chosen $t_\delta/T$ ratio consistently surpasses the baseline, underscoring the efficacy of our approach.

**Table 4: Ablation studies on the different noise schedules.**

| Training Iters | 300K | | 350K | | 500K | |
|---|---|---|---|---|---|---|
| | FID | sFID | FID | sFID | FID | sFID |
| ADM (Linear) | 3.73 | 4.87 | 3.53 | 4.78 | 3.43 | 4.34 |
| ADM-BS | **3.33** | **4.34** | **3.25** | **4.32** | **2.99** | **4.20** |
| ADM (Cosine) | 3.15 | 4.46 | 3.10 | 4.40 | 2.97 | 4.32 |
| ADM-BS | **2.96** | **4.16** | **2.94** | **4.16** | **2.90** | **4.14** |

**Robustness to Noise Schedule.** Since the convergence phenomenon remains unaffected by various noise schedules, we undertook a comparison involving different noise schedules (e.g., cosine and linear schedules) on the CIFAR-10 dataset, as outlined in Table 4. The results highlight ADM-BS's robustness against these diverse noise schedules. Notably, ADM-BS not only surpasses the baseline in terms of generation quality but also enhances training speed by a factor of 1.67× across both noise schedules. For more comparisons of different noise distribution (e.g., EDM [28]), please refer to Section 3.4 in the Appendix. The experimental results demonstrate that our method exhibits robustness across various noise schedules.

## 4.5 Qualitative Comparison

**Fast training.** In Figure 7, we present a comparison of samples generated by ADM and ADM-BS, trained across varying iterations (i.e., 50K, 100K, 200K, and 500K). ADM-BS can generate high-quality samples after only 100K iterations, achieving this at a much quicker rate than ADM. Moreover, the samples generated by ADM-BS exhibit a noticeably higher quality than those from ADM. This suggests that our method effectively improves the training efficiency.

**Unconditional generation.** In Figure 8, we compare the unconditional generation results of ADM and ADM-BS on the FFHQ and AFHQ-D datasets. It is observed that the samples generated by ADM exhibit color shift, whereas those generated by ADM-BS do not demonstrate color-shift issues. This result implies that ADM-BS generates higher-quality samples.

**Fine-grained dataset.** We conducted visual comparisons using the fine-grained dataset Stanford Cars. The results in Figure 9, reveal that the images produced via the ADM method manifest considerable deficiencies in detail, characterized by pronounced

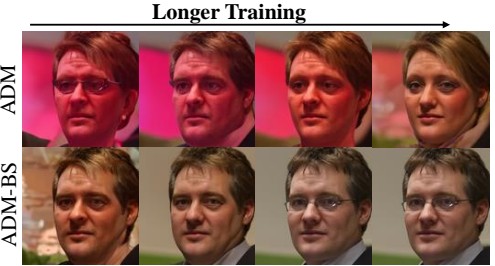

**Figure 7: The generation results of ADM and ADM-BS on the FFHQ dataset. Images in each column are sampled from 50K, 100K, 200K, and 500K training iterations.**

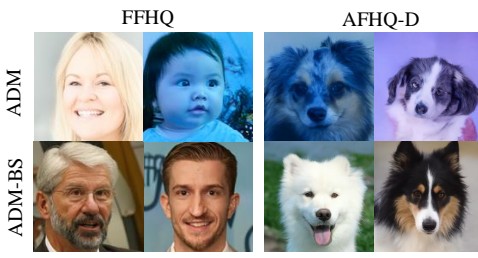

**Figure 8: Unconditional generation results.**

distortions and deformations, especially in the structural aspects of the generated cars. In contrast, the images generated using the ADM-BS demonstrate a remarkable enhancement in detail, with significantly reduced incidences of distortion or deformities.

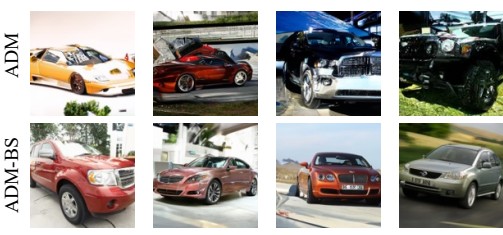

**Figure 9: Samples generated by ADM and ADM-BS method on Stanford Cars.**

## 5 Conclusion

In this paper, we investigate the forward diffusion process in diffusion models using optimal transport theory, proving an exponential decay of the upper bound of the Wasserstein distance between different distributions with time. Based on this theoretical insight, we design the DDPM-BS, a non-uniform timestep sampling method based on Bernoulli distribution. DDPM-BS aims to increase sampling at crucial intervals of timestep in the forward diffusion stage, significantly improving the training efficiency of the diffusion model. Our extensive experiments across popular generative benchmark datasets confirm the efficacy of DDPM-BS in speeding up diffusion model training and elevating the quality of the images generated. Furthermore, our approach enhances the performance of existing improved diffusion model training methods, which illustrates the generalization of our method.

# Acknowledgments

This work was supported in part by NSFC under Grant 61927809 and in part by STCSM under Grant 22DZ2229005.

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
