# OpenReview forum: "Non-uniform Timestep Sampling: Towards Faster Diffusion Model Training"
_acmmm.org/ACMMM/2024/Conference — MM2024 Poster_

### Official Review · Reviewer_CpHt · 2024-05-16

**Rating:** 6
**Confidence:** 3

**Summary:**

This paper introduces a non-uniform timestep sampling method to improve the training efficiency of diffusion models. This paper employs optimal transport theory to examine the forward process of the diffusion model, and find that the upper bound of the Wasserstein distance between any two specified moments exhibits an exponential reduction relative to their initial disparity. Based on this finding, this paper proposes a Bernoulli Distribution-Based sampling method for training diffusion models. Experiments show that the proposed method can accelerate diffusion model training.

**Strengths:**

1. Good empirical analysis combined with the optimal transport theory provides insights into the forward diffusion process.
2. The proposed method is both theoretically validated and empirically successful.
3. Comprehensive experiment results confirm the approach is effective.

**Limitations:**

1. Some theorems and propositions are hard to follow. Maybe better if authors can explain these concepts in a more easy-to-understand manner.
2. What are the differences between this work and other OT-based diffusion models (e.g., [1] [2] [3])? Is it correct that this work focuses on improving the efficiency of the diffusion sampling process while other OT-based diffusion models focus on improving the quality of generated images? It will be better if a short discussion can be included in the related works section.

References

[1] Gushchin N, Kolesov A, Korotin A, et al. Entropic neural optimal transport via diffusion processes. Advances in Neural Information Processing Systems, 2024, 36.

[2] Li Z, Li S, Wang Z, et al. DPM-OT: A New Diffusion Probabilistic Model Based on Optimal Transport. Proceedings of the IEEE/CVF International Conference on Computer Vision. 2023: 22624-22633.

[3] Wan Z Y, Baptista R, Boral A, et al. Debias coarsely, sample conditionally: Statistical downscaling through optimal transport and probabilistic diffusion models. Advances in Neural Information Processing Systems, 2024, 36.

**Suitability:**

3

---

### Official Review · Reviewer_hM1M · 2024-05-25

**Rating:** 3
**Confidence:** 3

**Summary:**

This paper introduces a non-uniform timestep sampling method leveraging optimal transport theory to accelerate diffusion model training while maintaining image quality.

**Strengths:**

The proposed non-uniform timestep sampling method is straightforward, easy to implement, and appears compatible with existing diffusion models.

**Limitations:**

1. Limited novelty: While the method is easy to use, its core concept might be too basic and lack significant innovation compared to existing techniques. It could be seen more as a tuning approach than a groundbreaking advancement.
2. Limited comparison and integration: The paper focuses solely on comparing the method to loss reweighting techniques. It would benefit from a more comprehensive evaluation that includes comparisons and potential integration with other state-of-the-art methods like consistency models, Rectified Flow, and Score Distillation Sampling.
3. Training efficiency concerns: The claimed faster training speed compared to the baseline ADM needs further clarification. I wonder if the number of samples per training iteration remains the same for both ADM and ADM-BS. Without this control, the observed speedup in the number of iterations might not be solely attributable to the proposed method. A more accurate comparison of training efficiency should focus on the actual training time, instead of training iteration.
4. This is a minor: Some equations in the paper lack numbering.

**Suitability:**

2

---

### Official Review · Reviewer_ioub · 2024-06-03

**Rating:** 4
**Confidence:** 2

**Summary:**

This paper introduces an innovative approach to accelerate the training of diffusion models. The optimal transport theory is leveraged to analyze the forward diffusion process, revealing that the state distribution of the diffusion model changes non-uniformly over time. Based on this insight, a non-uniform timestep sampling method is proposed using a Bernoulli distribution to concentrate on significant timesteps, thereby speeding up training and improving image generation quality.

**Strengths:**

1. **Theoretical Contribution.** The paper provides a rigorous theoretical analysis using optimal transport theory to understand the diffusion process, offering new insights into the non-uniform rate of change in the diffusion model's state distribution.
2. **Empirical Methodology.** The authors validate their theoretical findings through empirical experiments, demonstrating the practical effectiveness of their proposed non-uniform sampling method in reducing computational overhead and enhancing image quality.
3. **Novelty in Approach.** The non-uniform timestep sampling based on the Bernoulli distribution is a novel approach that contrasts with the uniform sampling methods traditionally used in training diffusion models.
4. **Potential Impact.** The proposed method is likely to be widely adopted by researchers and practitioners working on generative models due to its potential to significantly reduce training times and improve results without compromising quality.

**Limitations:**

1. **Relevance to ACM MM.** The method proposed only focuses on unimodal diffusion models. Diffusion-based models are prominent for multimodal generation tasks. If more experiments with multimodal settings are provided for illustration, this paper would be more suitable for ACM MM.
2. **Inconsistent Terminology.** The method proposed is called DDPM-BS before the Experiment section but is suddenly changed to ADM-BS from subsection 4.2.

**Suitability:**

2

---

### Official Review · Reviewer_CA4o · 2024-06-03

**Rating:** 2
**Confidence:** 3

**Summary:**

The paper proposes a non-uniform timestep sampling strategy for diffusion models, which can reduce training time and enhance the quality of generated images. This approach is based on the observation that the Wasserstein distance between the distributions of two timesteps decreases exponentially over time.

**Strengths:**

1) The observation that the Wasserstein distance between the distributions of two timesteps decreases exponentially is interesting and meaningful.
2) The theoretical part of the Wasserstein gradient flow is solid.
3) The proposed non-uniform timesteps indeed reduce the training time and improve the quality of the generated images.

**Limitations:**

1) In the Ablation Study, it is shown that performance improves as $p$ increases, with $p=0.8$ yielding the best results. What about $p=0.9$ or $p=1.0$? Does the diffusion model need to be trained on timesteps greater than $0.8T$? Additionally, when $\frac{t_\delta}{T}=0.8$, the proposed method is the same as uniform timestep sampling. As $\frac{t_\delta}{T}=0.8$, the probability density function for time sampling becomes $f(t)=\frac{1}{T}$​ for all timesteps. What is the difference between the proposed method and uniform timestep sampling? Figure 2 is confusing.

2) The connection between the analysis and the method appears weak. In Figure 1(b), the Wasserstein distance approaches 0 after 600 timesteps. However, as shown in Figure 6, the performance is best when $\frac{t_\delta}{T} = 0.8$. Could the author discuss this discrepancy?

3) The Wasserstein distance appears to depend on the noise schedule. How does the method perform on other types of diffusion models, such as VE-SDE[2]? Additionally, there is a lack of comparison with EDM[1].


[1] Karras, Tero, et al. "Elucidating the design space of diffusion-based generative models." Advances in Neural Information Processing Systems 35 (2022): 26565-26577.

[2] Song, Yang, et al. "Score-based generative modeling through stochastic differential equations." arXiv preprint arXiv:2011.13456 (2020).

**Suitability:**

3

---

### Meta-Review · Area_Chair_9xuo · 2024-06-30

**Recommendation:** Accept (Poster)
**Confidence:** 4

**Metareview:**

This paper introduces an approach to enhance the training efficiency of diffusion models by employing a non-uniform timestep sampling strategy based on the theory of optimal transport. While concerns, such as claims regarding non-uniform sampling given the choice of hyperparameters, were initially raised, after the rebuttal, all reviewers concur with the merits of the work. The ACs agree with this assessment and, therefore, recommend accepting the paper. Congratulations to the authors! Please incorporate these suggestions into the camera-ready version to enhance the paper's clarity and novelty and include new results from the rebuttal.